# New insights in photodynamic inactivation of *Leishmania amazonensis*: A focus on lipidomics and resistance

**Fernanda V. Cabral[1], Michela Cerone[2], Saydulla Persheyev[3], Cheng Lian[3], Ifor D. W. Samuel[3], Martha S. Ribeiro[1]\*, Terry K. Smith[2]\***

**1** Center for Lasers and Applications, Nuclear and Energy Research Institute (IPEN/CNEN), São Paulo, Brazil, **2** Schools of Biology & Chemistry, BSRC, University of St. Andrews, St Andrews, Fife, United Kingdom, **3** Organic Semiconductor Centre, School of Physics and Astronomy, University of St Andrews, St Andrews, United Kingdom

\* tks1@st-andrews.ac.uk (TKS); marthasr@usp.br (MSR)

**Data Availability Statement:** All relevant data are within the manuscript and its Supporting information files.

## Abstract

The emergence of drug resistance in cutaneous leishmaniasis (CL) has become a major problem over the past decades. The spread of resistant phenotypes has been attributed to the wide misuse of current antileishmanial chemotherapy, which is a serious threat to global health. Photodynamic therapy (PDT) has been shown to be effective against a wide spectrum of drug-resistant pathogens. Due to its multi-target approach and immediate effects, it may be an attractive strategy for treatment of drug-resistant *Leishmania* species. In this study, we sought to evaluate the activity of PDT *in vitro* using the photosensitizer 1,9-dimethyl methylene blue (DMMB), against promastigotes of two *Leishmania amazonensis* strains: the wild-type (WT) and a lab induced miltefosine-resistant (MFR) strain. The underlying mechanisms of DMMB-PDT action upon the parasites was focused on the changes in the lipid metabolism of both strains, which was conducted by a quantitative lipidomics analysis. We also assessed the production of ROS, mitochondrial labeling and lipid droplets accumulation after DMMB-PDT. Our results show that DMMB-PDT produced high levels of ROS, promoting mitochondrial membrane depolarization due to the loss of membrane potential. In addition, both untreated strains revealed some differences in the lipid content, in which MFR parasites showed increased levels of phosphatidylcholine, hence suggesting this could also be related to their mechanism of resistance to miltefosine. Moreover, the oxidative stress and consequent lipid peroxidation led to significant phospholipid alterations, thereby resulting in cellular dysfunction and parasite death. Thus, our results demonstrated that DMMB-mediated PDT is effective to kill *L. amazonensis* MFR strain and should be further studied as a potential strategy to overcome antileishmanial drug resistance.

**Funding:** CAPES 88887.364974/2019-00 Dr Fernanda V. Cabral Royal Society CH160144 Ifor D. W. Samuel CNPq #465763/2014-6 Dr Fernanda V. Cabral.

**Competing interests:** The authors have declared that no competing interests exist.

**Abbreviations:** Akt, protein kinase B; CDP-DAG, cytidine-diphosphate diacylglycerol; CL, cutaneous leishmaniasis; CTP, cytidine 5′-triphosphate; DMMB, 1,9-dimethyl methylene blue; FA, fatty acid; GPI, glycosylphosphatidylinositol; IPC, inositol phosphoceramide; LD, lipid droplets; MFR, miltefosine-resistant; MT, miltefosine transporter; PA, phosphatidic acid; PC, phosphatidylcholine; PDT, photodynamic therapy; PE, phosphatidylethanolamine; PG, phosphatidylglycerol; PI, phosphatidylinositol; PI3K, phosphatidylinositol 3-kinase; PL, phospholipid; PS, phosphatidylserine; ROS, reactive oxygen species; WT, wild-type.

# Introduction

Cutaneous leishmaniasis (CL) is a complex of diseases caused by multiple species of *Leishmania* protozoan parasites, transmitted by infected female phlebotomine sandflies [1]. CL has a worldwide distribution, and it has been estimated that over 600.000 to 1 million new cases occur annually [1, 2]. Several countries in the Americas, Mediterranean basin, Central Asia, and the Middle East account for nearly 95% of the global burden of CL [2].

The wide spectrum of clinical presentations ranges from a single nodule at the site of infection to disfiguring skin lesions [3]. It may also have a psychological impact on affected individuals, causing disabilities and social stigmatization [4, 5]. In more severe cases, lesions can be disseminated to other parts of the body either as ulcerative wounds or as multiple non-ulcerative nodules, which are commonly related to immunocompromised patients [1, 3].

Although the introduction of new drugs has brought several breakthroughs over CL chemotherapy, treatment options are still limited by their variable effectiveness over the different *Leishmania* species [6]. Daily injections of pentavalent antimonials for over a month are the current mainstay regimen of CL [6, 7]. Alternatively, amphotericin B can be applied as a second-line drug; however, it has also been based on a long-course intravenous regimen [1, 7]. Miltefosine, an anticancer drug, has emerged as a promising oral anti-leishmanial agent. However, it has only been approved for use in a few countries, as its actual therapeutic mode of action over CL is still unclear [7, 8]. In addition, because of long-term adverse effects, these treatments are often associated with systemic toxicity, resulting in poor adherence of patients to the strict therapeutic regimens [6, 7, 9]. This leads to the widespread misuse of drugs, and consequently may significantly contribute to the development of drug-resistant phenotypes in the field [10].

Moreover, given the wide variety of *Leishmania* species and different parasite-host interactions, variations in the sensitivity of these parasites to current medications have been widely reported [3, 10, 11]. Resistance to standard antileishmanial drugs has been a major concern over the past few years, as it poses as a serious public health threat [10]. Therefore, CL urgently needs more effective patient-compliant alternatives focused on non-toxic shorter regimens, preventing drug resistance and treatment failure [1, 12].

In this regard, improved strategies to prevent the emergence of resistant phenotypes have been proposed over the past decades [12]. On this basis, multiple target therapies hold promise for the treatment of CL, particularly for unresponsive or refractory cases [10].

Photodynamic therapy (PDT) has been shown to promote antimicrobial activity against several pathogens, including multidrug-resistant bacteria and some yeast species [13]. It is a light-based technology involving a light source with an emission spectrum that matches the absorption of a photosensitizer in the presence of oxygen [14]. The photochemical reactions induced by PDT may yield high amounts of reactive oxygen species (ROS) able to kill microbial cells [14]. Due to PDT multi-target features, ROS may reach lipids, proteins, and nucleic acids, thereby promoting a critical disruption of cellular homeostasis and functions [13, 14]. Thus, pathogens are less likely to develop resistance, which may give an advantage over single-target drugs. Additionally, PDT involves the use of a PS topically applied and light directly delivered to the target, which also may help patients to complete the whole course of treatment [15].

Several reports have shown the great efficiency of PDT to target different *Leishmania* species using a wide range of photosensitizers [16–18]. Among them, phenothiazine-based dyes have shown to be attractive for their low cost and generating large amounts of ROS over a short period, promoting immediate effects. In addition, they absorb light in the red region of the spectrum, and the light that is not absorbed could be scattered forward and penetrate deeper into tissue than shorter wavelengths [19, 20].

Recently, we have reported that the phenothiazine dye 1,9-dimethyl methylene blue (DMMB) was highly effective against different *Leishmania* species without toxicity to mammalian cells [21, 22]. Therefore, aiming to further assess the potential of this promising photosensitizer, we evaluated for the first time the susceptibility factors involving DMMB-mediated PDT (DMMB-PDT) against promastigotes of *L. amazonensis* using the wild-type (WT) and miltefosine-resistant (MFR) strains. In our current study, parasites were exposed to DMMB-PDT and various cellular processes were investigated to give insight into the possible underlying mode of action. These included oxidative stress conditions, as assessed by detection of intracellular levels of ROS; mitochondrial potential and lipid body content as determined by immunofluorescence staining; and quantitative lipidomics analysis using electrospray-mass spectrometry.

## Experimental section

### Light source and photosensitizer

Here we used a red LED (660 nm peak wavelength, 25 nm full width at half maximum, $660 \pm 12.5$ nm) at an irradiance of 20 mW/cm$^2$. The light source illuminates the wells from underneath, with distance of 4 mm from the bottom of the well. Two light doses were used: i) lethal condition (50 J/cm$^2$, t = 2500 s); ii-) sublethal condition (8 J/cm$^2$, t = 400 s). The absorbance of the photosensitizer 1,9-dimethyl methylene blue (DMMB) was measured using a plate-reader (SpectraMax 83 M4, Molecular Devices, USA).

### Parasites

*L. amazonensis WT* (MHOM/BR/73/M2269) promastigotes were grown at 28˚C in M199 medium (Sigma-Aldrich) supplemented with 10% heat-inactivated fetal bovine serum (FBS; Gibco™ Invitrogen Corporation), 40 mM HEPES pH 7.4 (Sigma-Aldrich), 2.5 mg/mL hemin (Sigma-Aldrich) and 10 mM Adenosine (Sigma-Aldrich) [23]. *L. amazonensis* MFR was selected from the reference strain M2269 (MF 150.3–1 line) as previously reported [24]. MFR parasites were grown in the same media as WT, but in the presence of 150 μM of miltefosine (Sigma-Aldrich).

### Miltefosine and DMMB-PDT activity against WT and MFR *L. amazonensis* promastigotes

Miltefosine activity for the WT and MFR *L. amazonensis* promastigotes was assesed by the addition of serial dilutions (dilution factor 1:2) of 100 μL of MF (0–500 μM) into a 96-well plate. Then, 100 μL of *Leishmania* promastigotes were seeded at a density of 1×10$^6$ per well to obtain a final volume of 200 μL. Miltefosine was incubated for 24 h at 28˚C.

PDT activity against both strains was determined by the addition of serial dilutions (dilution factor 1:2) of 100 μL of DMMB (0–3000 nM) into a 96-well plate. Parasites were then seeded at 1×10$^6$ per well in a final volume of 200 μL. Before irradiation, DMMB was incubated for 10 min (pre-irradiation time) to allow the photosensitizer uptake. Then, cells were irradiated using the red LED for 2500 s to deliver a dose of 50 J/cm$^2$. Untreated parasites were used as a negative control in a different plate. Incubation of parasites with varying concentrations of DMMB without light was also assessed to evaluate the cytotoxicity of the photosensitizer in the dark.

For both assays, cell viability was assessed after treatment (24 h for miltefosine and directly after DMMB-PDT). Briefly, 10 μL of a stock solution of resazurin (1.1 mg/mL) (Alamar blue, Sigma-Aldrich) was added to each well and incubated for 5 h at 28˚C. Fluorescence intensity

was determined by using a plate reader (Gen5 Reader, BioTek) at $\lambda_{ex}$ = 530 nm and $\lambda_{em}$ = 590 nm. The half-maximal effective concentration ($EC_{50}$) was obtained by sigmoidal regression analysis using GraphPad Prism 7.0 software.

## Detection of reactive oxygen species (ROS)

ROS production was measured using the indicator 2'-7'-dichlorodihydrofluorescein diacetate (DCFH-DA) (Abcam), a dye that evaluates total ROS in live cells [23]. Both *Leishmania* strains at a density of $1\times10^6$ per well were seeded into 96-well plates. Cells were treated with DMMB-PDT at 750 nM, delivering a light dose of 50 J/cm$^2$. Parasites were also incubated in the presence of DMMB at 750 nM without light. Untreated cells were used as a negative control and 25 μM $H_2O_2$ as a positive control. After treatment, 2 μL of a stock solution of DCFH-DA (1 mM) was added to each well and incubated for 45 min. Fluorescence intensity was determined using a plate reader (Gen5 Reader, BioTek) at $\lambda_{ex}$ = 485 nm and $\lambda_{em}$ = 530 nm.

## Immunofluorescence microscopy

For immunofluorescence, $1\times10^6$ parasites per well were seeded into 96-well plates and exposed to DMMB-PDT (8 J/cm$^2$, DMMB 750 nM). The light dose was reduced in order to understand the role of DMMB-PDT on parasites. Cells were centrifuged at 1400 g for 10 min and gently washed in 1 ml PBS, followed by fixation with 4% paraformaldehyde for 20 min at room temperature. Afterward, cells were allowed to adhere to poly-L-lysine coated slides before staining with i-) DAPI: cells were fixed immediately after DMMB-PDT and stained with DAPI (4,6-diamidino-2-phenylindole) for 5 min (2 μg/ml, diluted in PBS), in the dark, at room temperature; ii-) Nile red: the Nile red (Sigma-Aldrich) accumulation was assessed in two different moments: Directly after, and 1h after DMMB-PDT. Cells were fixed and incubated with 10 μg/ml Nile red, in the dark, at room temperature, for 30 min and DAPI for 5 min; (iii) Mito-Tracker CMXRos: For mitochondrial labeling, cells were treated and incubated with Mito-Tracker CMXRos (Invitrogen) at 28˚C, in a final concentration of 100 nM, for 2h. Experiments were also performed at two different time points: Immediately and 1h after DMMB-PDT. Then, parasites were fixed and allowed to sediment and adhere to slides before staining with DAPI. All images were acquired with a fluorescence microscope (DeltaVision Imaging System) connected to a digital camera system and processed by softWoRx image analysis software.

## Lipid analysis

WT and MFR *L. amazonensis* promastigotes were harvested at a density of $1\times10^8$ cells/ml and exposed to DMMB-PDT (8 J/cm$^2$, DMMB 750 nM). After that, total lipid extraction was performed according to Bligh and Dyer method [25]. Cells were collected by centrifugation (800 x g, 10 min), washed, and resuspended with 100 μL of PBS and transferred to a glass tube. Then, 375 μL of chloroform:methanol 1:2 (v/v) were added and vortexed. Ten-μL of lipid internal standards SPLASH LIPIDOMIX Mass Spec Standard (Avanti Polar Lipids) were added to each sample followed by intense agitation for at least 15 min. Thereafter, 125 μL of chloroform was added and vortexed. Then, 125 μL of $H_2O$ was added and vortexed again to make the samples biphasic. Samples were centrifuged at room temperature (1000 g, 5 min) and the lower phase (organic) was transferred to a new glass tube, dried under nitrogen, and stored at 4˚C.

Before the electrospray ionization-tandem mass spectrometry (ESI-MS-MS) analysis, 15 μL of 1:2 (v/v) chloroform:methanol and 15 μL of acetonitrile:isopropanol:water (6:7:2) were added to the sample, and lipids were resuspended. Phospholipids were analyzed using Absceix 4000 QTrap, a triple quadrupole mass spectrometer with a nanoelectrospray source.

Lipid analysis was determined in positive and negative ion modes using a capillary voltage of 1.25 kV. Tandem mass spectra scanning (MS/MS) (daughter, precursor, and neutral loss scans) were assessed using nitrogen as the collision gas, with collision energies between 35 and 90 V as previously described [26]. Each spectrum (m/z 600–1000) includes at least 50 repetitive scans. Glycerophospholipids, including phosphatidic acid (PA) and phosphatidylglycerol (PG), were detected by precursor scanning for m/z 153 in negative ion mode. Phosphatidylcholine (PC) in positive ion mode, parent-ion scanning of m/z 184. PI/IPC (phosphatidylinositol/inositol-phosphorylceramide) was detected in negative ion mode by parent-ion scanning of m/z 241. Phosphatidylethanolamine (PE) in negative ion mode, parent-ion scanning of m/z 196 and phosphatidylserine (PS) in negative ion mode, neutral loss scanning of m/z 87.

Identification of phospholipids was based upon previous *Leishmania* lipidomic analyses [27, 28] and LIPID MAPS Lipidomics Gateway (http://www.lipidmaps.org). The concentration of each phospholipid class was obtained based on the corresponding internal standard from the SPLASH LIPIDOMIX Mass Spec Standard (Avanti Polar Lipids).

## Statistical analysis

Data were obtained in triplicate. ROS production was analyzed by two-way analysis of variance (ANOVA) followed by the Bonferroni post-test. Lipid analysis was performed by one-way analysis of variance (ANOVA) followed by Tukey post-test. Differences were considered statistically significant when $p < 0.05$.

## Results

### DMMB absorption spectra

DMMB presents an absorbance band between 500 and 700 nm with two peaks around 542 and 647 nm. A red LED emitting $\lambda = 660 \pm 12.5$ nm was therefore used to match the absorption of the photosensitizer (Fig 1).

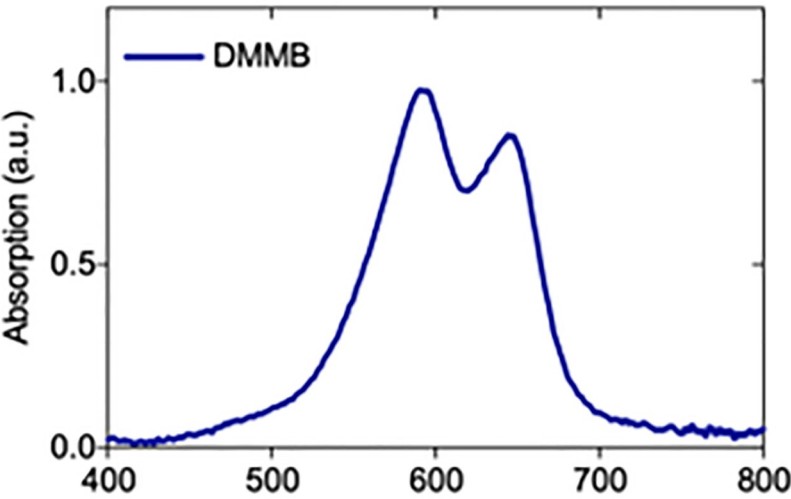

**Fig 1. DMMB absorption spectra.**

## MFR promastigotes are equally susceptible to DMMB-PDT as the WT strain

Miltefosine activity against both WT and MFR strains was determined by the incubation of parasites in the presence of increasing concentrations of miltefosine (0–500 μM). The miltefosine concentration required to achieve a reduction of 50% ($EC_{50}$) of MFR parasites was 5.5-fold higher (140.2 ± 3.6 μM) compared to the WT (25.5 ± 0.22 μM) strain (Fig 2A). DMMB-PDT was also assessed by varying the concentrations of DMMB (0–3000 nM), in which $EC_{50}$ value was found to be 1.5-fold lower on MFR (34.6 ± 0.12 nM) compared to the WT (53.0 ± 0.11 nM). DMMB-PDT was effective in killing both strains of parasites. Indeed, both lines were equally susceptible to DMMB-PDT-induced oxidative stress (Fig 2B). Additionally, our previous studies showed that the photosensitizer at high concentrations without light did not promote any cytotoxic effect to the parasites [21].

## DMMB-PDT produces higher levels of intracellular ROS on MFR *L. amazonensis* strain

To measure the redox state of cells, ROS detection was determined directly after DMMB-PDT. The H2DCFDA probe is a sensitive method used to detect minor changes in the intracellular redox system promoted by oxidative stress. Therefore, fluorescence of H2DCFDA is proportionally related to total ROS generation, which was identified by an increase in fluorescence signal, as observed in Fig 3. Although the fluorescence is measured in arbitrary units (AU) the measurement conditions were kept the same so that the fluorescence intensity can be used to give information about the relative amount of ROS. Low levels of ROS were produced over the WT and MFR untreated control (2739.7 ± 418.9; 2716 ± 175.8 AU, respectively). Moreover, DMMB 750 nM (without light) had similar values (2923.6 ± 684.1 for WT; 3630.3 ± 60.9 AU for MFR) with no significant differences between strains. The positive control group ($H_2O_2$ at 25 μM) showed a ~4-fold and ~6-fold increase in ROS production over the WT (12.182 ± 121.6 AU) and MFR (16.118.3 ± 693.8 AU) strains compared to the negative control, respectively. Thus, the levels of ROS detected for the MFR strain were almost a third higher

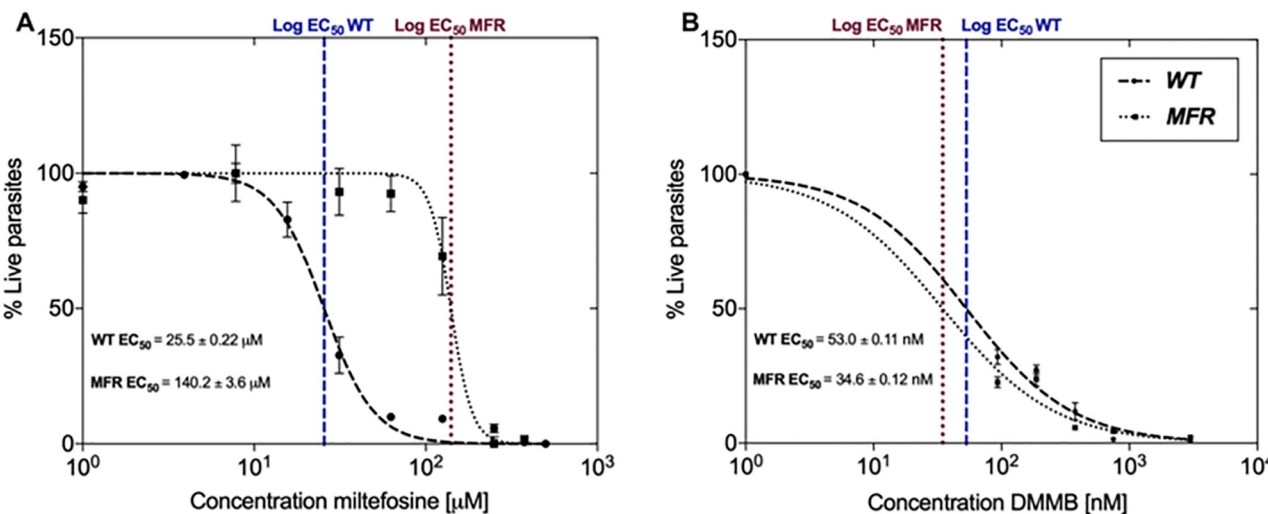

**Fig 2. Susceptibility of WT and MFR *L. amazonensis* promastigotes to (A) miltefosine at increasing concentrations (0–500 μM) and (B) DMMB-PDT.** Parasites were exposed to PDT at 50 J/cm² and different concentrations of DMMB (0–3000 nM). Values represent mean ± SD.

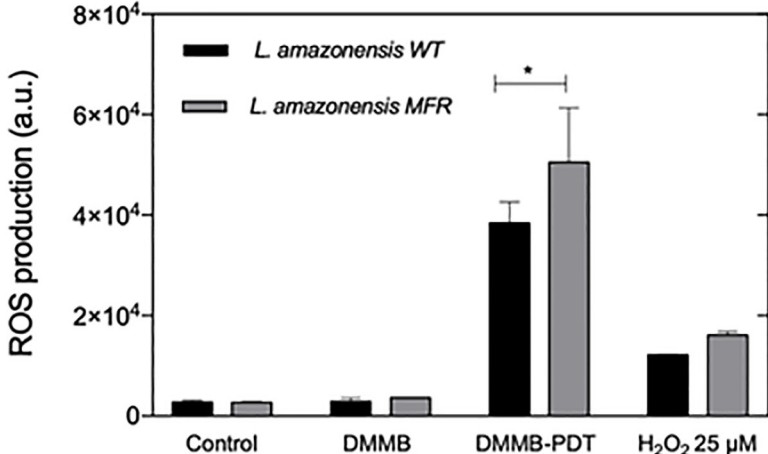

**Fig 3. Total ROS production using the fluorescent probe H2DCFDA in WT and MFR *L. amazonensis* promastigotes.** Parasites were treated with DMMB-PDT at 50 J/cm² and DMMB at 750 nM. Untreated parasites were used as a negative control and $H_2O_2$ at 25 μM as a positive control. The levels of ROS produced only by DMMB at 750 nM was also assessed. ROS production was analyzed by two-way analysis of variance (ANOVA) followed by the Bonferroni post-test. The values shown represent the mean ± SD. * denotes statistically significant differences between strains when $p < 0.05$.

than for the WT. We observed large increases in ROS following DMMB-PDT. For the WT we observed a fluorescence intensity of (38.410 ± 4202 AU) which is 14.6 times higher than the untreated control. For the MFR strain we observed even higher fluorescence intensity of (50.489 ± 10.856 AU) which is 18.5 times higher than the untreated control. The fluorescence intensity also shows that the MFR strain produces 24% more ROS than the WT strain.

## DMMB-PDT does not affect DNA-containing organelles of promastigotes

We also investigated any immediate changes in the DNA of parasites due to DMMB-PDT. Cells were fixed and stained with DAPI (kinetoplast—mitochondrial DNA, and nucleus) directly after DMMB-PDT treatment, followed by image analysis using a high-resolution fluorescence microscope. Results show that both DNA-containing organelles were well stained in all groups, resulting in a high-intensity signal with bright areas, mainly over kinetoplast. Indeed, no differences in staining were observed between both controls and DMMB-PDT groups. Both treated cell lines revealed the same structures of a kinetoplast (bar-shaped) and nucleus (round) as untreated controls. Under these conditions, parasites did not differ in their morphological phenotype, including shape and size. No signs of DNA degradation, nuclear condensation, and/or fragmentation were observed (suppl Fig 1 in S1 File).

## DMMB-PDT promotes mitochondrial dysfunction on both WT and MFR strains

Mitochondria of WT and MFR parasites were visualised by incubating with MitoTracker red (at 100 nM) for 2 h at two different time points: i-) Directly after, and ii-) 1h after DMMB-PDT.

The general cell, morphology, i.e. elongated cell body and long flagellum extending out of the flagellar pocket were preserved in all groups for both time points. In untreated controls, the dye was well retained after fixation, displaying a bright and well-defined red color over the single large mitochondrion across the entire cell, here defined as "WT control D" and "MFR

control D" (Fig 4A and 4E). However, when exposed to DMMB-PDT, the loss of membrane integrity resulted in a reduced signal even directly after treatment (defined as WT DMMB-PDT D and MFR DMMB-PDT D), as shown in Fig 4B and 4F.

To further investigate the mitochondrial membrane potential at a later time course, Mito-Tracker staining was assessed 1h post-treatment. As a result, the same pattern was observed

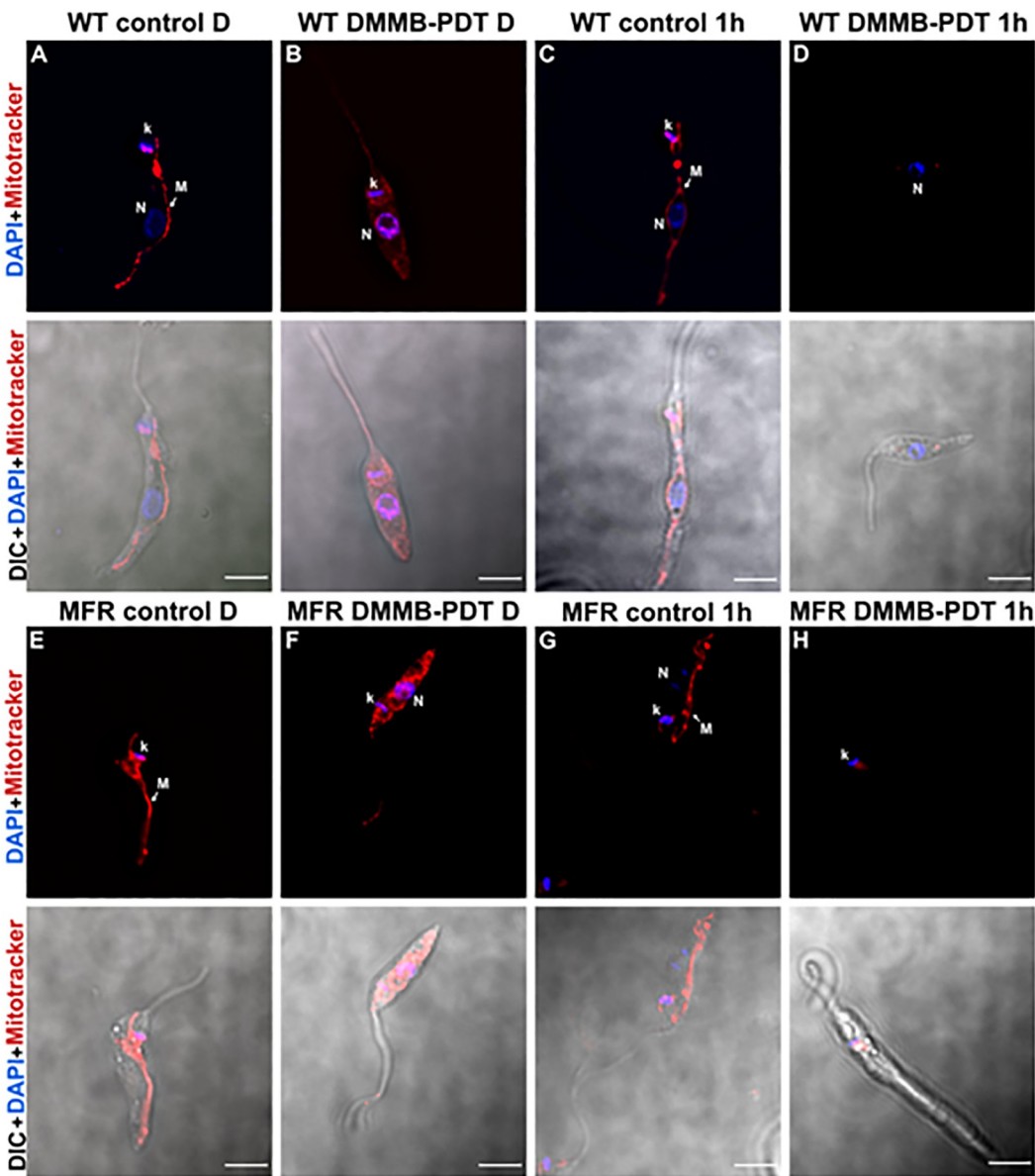

**Fig 4. Differential interference contrast (DIC) and immunofluorescence staining images of WT and MFR *L. amazonensis* promastigotes treated with DMMB-PDT at 8 J/cm² in the presence of 750 nM of DMMB.** A) Refers to WT untreated control stained directly after DMMB-PDT (WT control D). B) Refers to WT exposed to treatment and stained directly after DMMB-PDT (WT DMMB-PDT D). C) Refers to WT untreated control stained 1h after DMMB-PDT (WT control 1h). D) Refers to WT exposed to treatment and stained 1h after DMMB-PDT (WT DMMB-PDT 1h). E) Refers to MFR untreated control stained directly after DMMB-PDT (MFR control D). F) Refers to MFR exposed to treatment and stained directly after DMMB-PDT (MFR DMMB-PDT D). G) Refers to MFR untreated control stained 1h after DMMB-PDT (MFR control 1h). H) Refers to MFR exposed to treatment and stained 1h after DMMB-PDT (MFR DMMB-PDT 1h). Nuclei were stained with DAPI (blue fluorescence) and mitochondria were stained with Mito tracker red (red fluorescence) directly after PDT. N = Nuclei; k = Kinetoplast; M = Mitochondrion. Scale bar = 5 μm.

for untreated control (here defined as WT control 1h and MFR control 1h), and parasites maintained their morphological features (Fig 4C and 4G). However, unlike the first time point (directly after DMMB-PDT), no red fluorescent signal was detected for treated groups, suggesting a loss of mitochondria membrane potential, thus sustained mitochondrial dysfunction over the 2-hour labeling period. DMMB-PDT treated cells (defined as WT DMMB-PDT 1h and MFR DMMB-PDT 1h) only showed a low light intensity, barely detectable (Fig 4D and 4H). We also observed that there were no significant changes in the parasite's nucleus or kinetoplast, as detected by DAPI staining, after the labeling period.

## DMMB-PDT increases cytoplasmic lipid droplets of both WT and MFR phenotypes

The presence of intracellular lipid droplets (LDs) was investigated by staining with Nile Red either directly after DMMB-PDT, or 1 h after DMMB-PDT. Subsequent fluorescence microscopy followed by image analysis shows a few small LDs were observed in untreated control groups of both cell lines, as expected under physiological conditions. For both untreated controls stained directly (WT control D, MFR control D) or 1h after DMMB-PDT, (WT control 1 h and MFR control 1h) the red points with low intensity over specific areas along the parasites' cytoplasm show the spherical shape of LDs (Fig 5A, 5C, 5E and 5G).

However, the cellular stress in DMMB-PDT treated parasites, both WT and MFR, caused an immediate and significant increase in the stained LD as well as LDs fusion, which is shown by the greater fluorescence intensity diffused across the cytosol (Fig 5B and 5F). Indeed, 1 hour after DMMB-PDT (defined as WT DMMB-PDT 1h and MFR DMMB-PDT 1h) there was an abnormal accumulation of LDs in both strains, resulting in some large-sized structures, as shown in Fig 5D and 5H.

## Lipidomics analysis reveals a different lipid profile between the WT and MFR strains

The lipidomics profile was evaluated using a triple quadrupole mass spectrometer fitted with a nano-electrospray source. Spectra were obtained from the total lipid extracts and evaluated in positive and negative ion mode for both WT and MFR *L. amazonensis*. As expected many of the various glycerophospholipid species were observed as well as inositol-phospoceramide, in keeping with previous lipidomic analyses of other *Leishamania* species [23, 28–32]. Phospholipid identities with the corresponding mass over charge (*m/z*), lipid components, and peaks obtained from high resolution survey scans are specified in Figs 6 and 7 and Table S1 in S1 File. After normalization using non-natural internal standards, the identified lipid species, and their amounts in the various samples were calculated and are represented in a heat-map (Fig 7). The molecular species that showed statistically significant differences were plotted in bar graphs according to the corresponding phospholipid (PL) classes (Fig 6 and S2-S6 Figures in S1 File).

ES-MS-MS negative ion spectra of inositol-containing PLs revealed some differences over phosphatidylinositol (PI) molecular species and minor changes of inositol-phosphorylceramide (IPC). Precursor ion scanning for *m/z* 241 identified that two of the most abundant PI species; 851 and 782 *m/z* (PI a-36:0) and (PI 30:0) were 2.2-fold and 1.6-fold higher for MFR than for the WT strain (Figure S2B and S2E in S1 File). Interestingly, for the *m/z* 935 (PI 42:8), one of the species produced in lesser amounts in WT, the abundance was found to be 3.4-fold higher for MFR (Fig 6D). While many of the other PI and IPC species showed no significant differences in their relative amounts (Figs 6B, 6C and 7 and S2A, S2C, S2D, S2F and S2G Figure in S1 File).

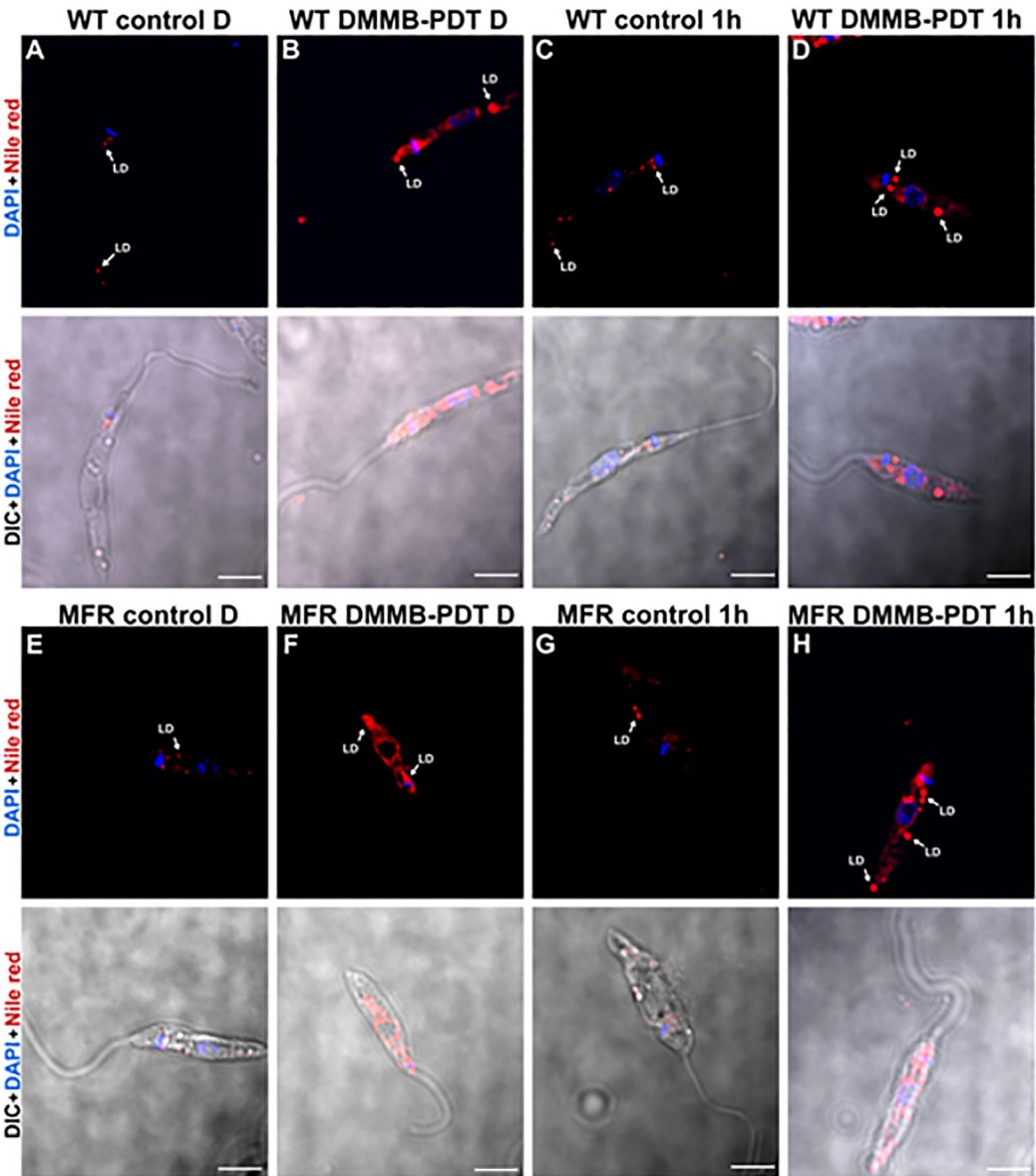

**Fig 5. Differential interference contrast (DIC) and Nile Red and DAPI immunofluorescence staining, representative images of WT and MFR *L. amazonensis* promastigotes treated with PDT at 8 J/cm² in the presence of 750 nM of DMMB.** A) Refers to WT untreated control stained directly after DMMB-PDT (WT control D). B) Refers to WT exposed to treatment and stained directly after DMMB-PDT (WT DMMB-PDT D). C) Refers to WT untreated control stained 1h after DMMB-PDT (WT control 1h). D) Refers to WT exposed to treatment and stained 1h after DMMB-PDT (WT DMMB-PDT 1h). E) Refers to MFR untreated control stained directly after DMMB-PDT (MFR control D). F) Refers to MFR exposed to treatment and stained directly after DMMB-PDT (MFR DMMB-PDT D). G) Refers to MFR untreated control stained 1h after DMMB-PDT (MFR control 1h). H) Refers to MFR exposed to treatment and stained 1h after DMMB-PDT (MFR DMMB-PDT 1h). Nuclei were stained with DAPI (blue fluorescence) and lipid droplets were stained with Nile red (red fluorescence) directly after PDT. LD = Lipid droplet. Scale bar = 5 μm.

Negative ion mode using the precursor scanning for *m/z* 196 allowed us to identify ethanolamine-containing PLs from total lipids extracted from WT and MFR. As well as diacyl PE species, the presence of ether (phosphatidylethanolamine) PE (alkylacyl PE and alkenylacyl PE), i.e. plasmalogens were detected (Figs 6 and 7 and S4 Figure in S1 File). The overall PE

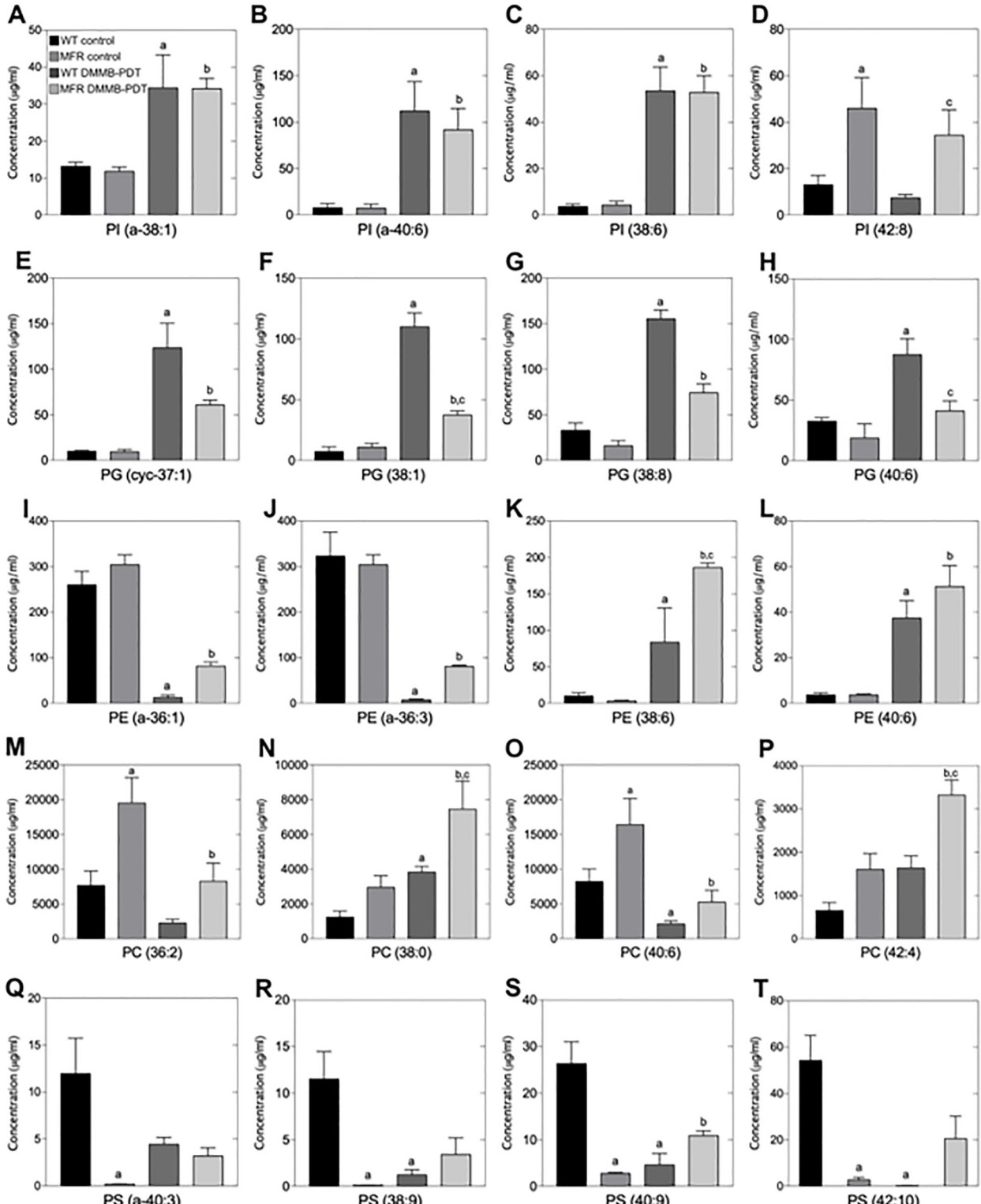

**Fig 6. Phospholipid analysis of WT and MFR _L. amazonensis_ promastigotes treated with PDT at 8 J/cm² in the presence of 750 nM of DMMB.** A-T represent the concentrations of some molecular species of the corresponding untreated and treated WT and MFR strains. PL class (IPC, PI, PA, PG, PE, PC and PS) was analyzed and quantified according to its corresponding internal "SPLASH" standard. "a" denotes statistically significant differences of PLs species compared to WT control. "b" denotes statistically significant differences of PLs species compared to MFR control. "c" denotes statistically significant differences of PLs species between WT DMMB- PDT and MFR DMMB-PDT.

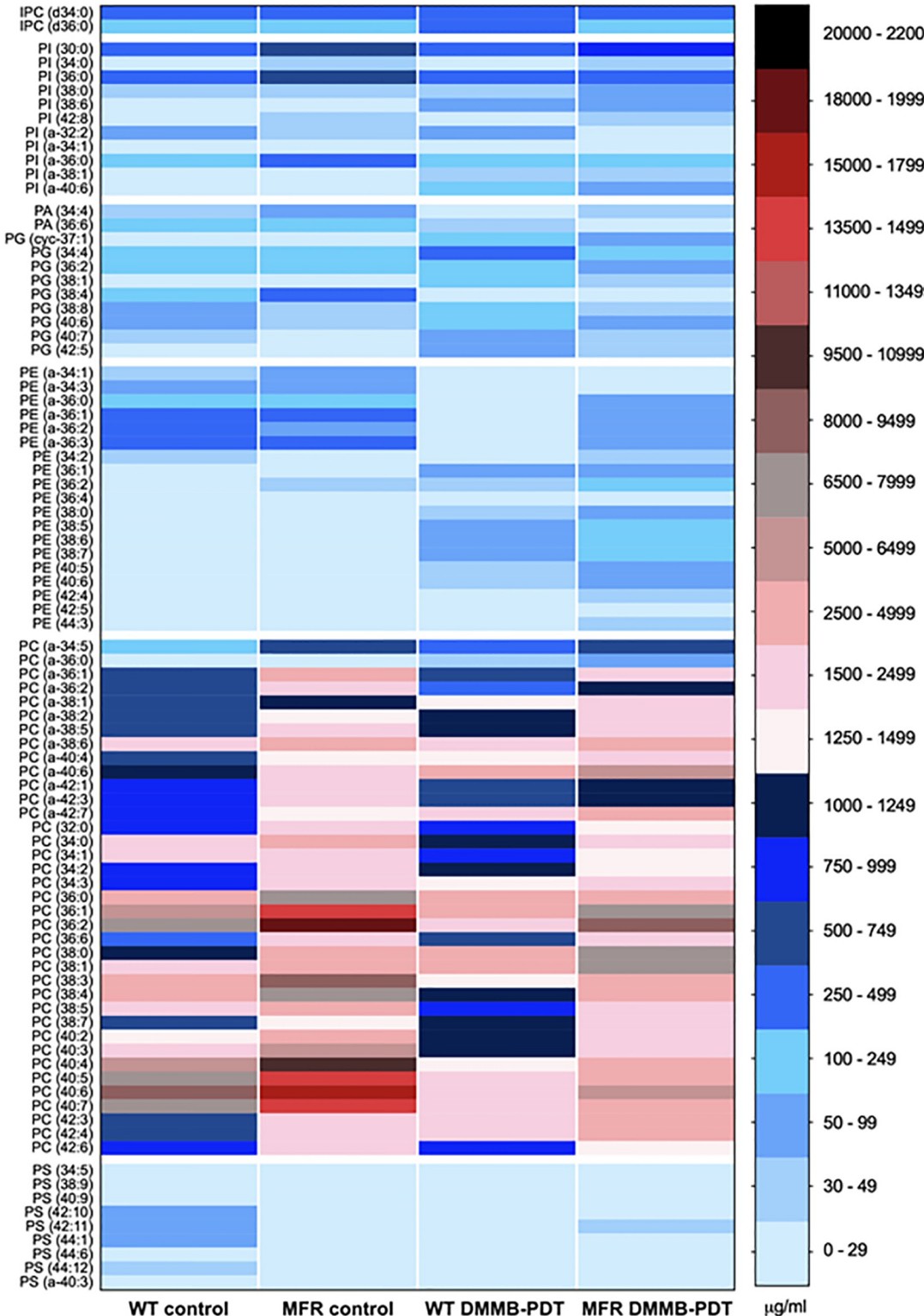

**Fig 7. Phospholipid analysis of WT and MFR *L. amazonensis* promastigotes treated with DMMB-PDT at 8 J/cm² in the presence of 750 nM of DMMB.** Bar on the right side refers to a color scale of the concentration (µg/ml) of each molecular species related to the corresponding PL class (IPC, PI, PA, PG, PE, PC and PS). Each PL class was analyzed and quantified according to its corresponding internal "SPLASH" standard.

composition did not show major differences in the relative abundance of this PL class between WT and MFR. However, one of the most abundant plasmalogen PE species *m/z* 728 (PE a-36:2) showed significantly lower levels in MFR, about 4 times compared to the WT. In contrast, *m/z* 701 (PE a-34:1), which is less abundant in the WT, is increased by 26% in MFR parasites (Fig 7 and S4 Figure in S1 File).

Precursor ion scanning for *m/z* 153 in negative ion mode can also reveal the phosphatidic acid (PA) and phosphatidylglycerol (PG) species and their relative amounts. The dynamic pool of PA species is synthesized in small numbers for both strains. Although they seem to be present in increased levels for MFR, but there is no statistically significant difference. PG is also normally a minor class of PLs and its synthesis follows the same trend as PA, with no relevant differences between the WT and MFR, even though some molecular species are present in smaller abundance on MFR parasites (Figs 6E–6H and 7 and S3 Figure in S1 File).

The PLs containing choline-based head groups were obtained in positive ion mode by precursor scanning for *m/z* 184. [M-H$^+$] ions of phosphatidylcholine (PC) revealed the presence of the main PLs subclasses, such as diacyl and alkylacyl (plasmanyl).

Lipid analysis showed that PC comprises the most abundant class of PLs in WT *L. amazonensis*. However, the differences between both strains revealed that the overall amounts of PC are substantially increased in MFR *L. amazonensis*, pointing out to 24 molecular species with higher amounts than in the WT strain. The main PC peaks includes *m/z* 734 (PC 32:0), *m/z* 760 (PC 34:1), *m/z* 785 (PC 36:2), *m/z* 788 (PC 36:1), *m/z* 811 (PC 38:4), *m/z* 812 (PC 38:3), *m/z* 832 (PC 40:7), *m/z* 834 (PC 40:6), *m/z* 861 (PC 42:6) (Figs 6M–6P and 7 and S5 Figure in S1 File).

Except for *m/z* 760, which was 1.6-fold increased, all other choline-containing PLs mentioned above doubled in PC abundance on MFR strain (Fig 7 and S5M Figure in S1 File). To be more exact, it was observed a 2.3-fold increase for the *m/z* 733, 811, 812, 861 PC species (Fig 7 and S5L, S5W and S5V, S5Ff Figure in S1 File). Levels of PC contents were 2.5 times higher for the *m/z* 785 and 788 in MFR (Fig 7 and S5Q and S5R in S1 File). Other PC molecular species produced in lower quantities were also detected at higher levels on the MFR phenotype. There was a 1.6-fold increase for the *m/z* 758 (PC 34:2), while the *m/z* 756 (PC 34:3), *m/z* 790 (PC 36:0), *m/z* 808 (PC 38:5), and *m/z* 837 (PC 40:4) species doubled in abundance for MFR, with statistically significant differences compared to the WT (Fig 7 and S5N-S5P, S5X in S1 File). It was observed that *m/z* 738 (PC a-34:5) and *m/z* 839 (PC 40:3) PC levels were 2.5-fold higher in MFR (Fig 7 and S5A and S5Z in S1 File). The PLs showing the peaks of *m/z* 772 (PC a-36:2) and *m/z* 777 (PC 36:6) had the greatest increase, resulting in a relative abundance 3.4 and 5.1 times higher for MFR than for the WT, respectively (Fig 7 and S5D and S5S Figure in S1 File).

Abundance of ether linked PC species was also significantly greater for MFR resulting in a doubled amount of the species *m/z* 734 (PC 32:0), *m/z* 792 (PC a-38:6), *m/z* 794 (PC a-38:5), while *m/z* 774 (PC a-36:1) and *m/z* 776 (PC a-36:0) demonstrated a 4.8-fold and 7.7-fold increasing over the WT strain (Fig 7 and S5B, S5C, S5G, S5H and S5L Figure in S1 File).

PS analysis was carried out in negative ion mode by scanning a neutral loss of *m/z* 87. The overall PS content revealed that this PL class composes a small percentage of total PLs in WT *L. amazonensis*. Surprisingly, the PS composition of MFR showed a significant reduction in the levels compared to the WT. MFR line revealed decreased amounts in PS series (about 99%) for the peaks with values of *m/z* 801 (PS 38:9), *m/z* 828 (PS a-40:3), *m/z* 853 (PS 42:11), *m/z* 855 (PS 42:10), *m/z* 903 (PS 44:1) (Fig 6 and S6A, S6B, S6D-S6F Figure in S1 File). It was observed around 95% decreasing for *m/z* 892 (PS 44:6), *m/z* 880 (PS 44:12), while the abundance of *m/z* 829 (PS 40:9) detected was 90% less for MFR (Fig 7 and S6C, S6G and S6H Figure in S1 File).

## DMMB-PDT promotes a rapid lipid remodeling of WT and MFR strains

Phospholipids were also analyzed after DMMB-PDT of both strains by ESI-MS/MS, allowing the identification of several changes in phospholipid levels and thus presumably their metabolism. Negative ion mode using the precursor scanning for *m/z* 241 revealed that in the WT DMMB-PDT group there was a ~40% overproduction of *m/z* 808 (IPC 36:0) compared to untreated WT, while this difference was not observed in the MFR cells upon DMMB-PDT (Fig 7 and S2A Figure in S1 File). We also observed that the levels of several PI species increased (~2.5 fold) for both strains after DMMB-PDT, i.e. *m/z* 878 (PI a-38:1), *m/z* 897 (PI a-40:6), while *m/z* 883 (PI 38:6) species had a 14.4-fold (WT DMMB-PDT) and 12.7-fold (MFR DMMB-PDT) (Fig 7 and S2C, S2D, S2G Figure in S1 File).

The overall lipid alterations detected through negative mode ion scanning for *m/z* 153 revealed a downward trend in PA abundance for both DMMB-PDT-treated parasites (Fig 7). In contrast, PG content was considerably enhanced after treatment in both strains. However, statistically significant increases were found in 6 species for WT DMMB-PDT, by 2.7-fold for *m/z* 821 (PG 40:7) and *m/z* 822 (PG 40:6). While a 4.7-fold and 6.1-fold increase for *m/z* 789 (PG 38:8) and *m/z* 853 (PG 42:5), respectively (Fig 7 and S3E-S3H Figure in S1 File). Impressively, there were two species of which a remarkable increase by 12.3-fold and 14.9-fold was detected (*m/z* 759 –PG cyc-37:1, *m/z* 805 –PG 38:1, respectively) after DMMB-PDT (Fig 7, and S3A, S3B Figure in S1 File).

Although the overall abundance of PG seems to be higher over MFR DMMB-PDT, only 3 species were significantly different from their own control. As mentioned earlier, while *m/z* 805 (PG 38:1) and *m/z* 759 (PG cyc-37:1) showed the largest increase over WT DMMB-PDT, a minor increase was detected for the same species on MFR DMMB-PDT (by 3.3-fold and 6.4-fold, respectively) (Fig 7, and S3A, S3B Figure in S1 File). Regarding *m/z* 789 (PG 38:8), both treated groups presented a similar 4.7-fold increase (Fig 7, and S3E Figure in S1 File). In addition, a statistically significant reduction of 90% was detected in *m/z* 798 (PG 38:4) for both treated strains.

In terms of PE, we found significant changes over 18 PE lipid contents. Unlike the other two PL classes mentioned earlier, 5 PE species were consistently decreased in both treated groups. Particularly, after DMMB-PDT, three major PE components, *m/z* 726 (PE a-36:3), *m/z* 728 (PE a-36:2) and *m/z* 729 (PE a-36:1) were reduced in 97.6%, 97.9% and 95.1% for the WT, while for MFR there was a reduction of 73.2%, 16.9% and 73.1% over the same Pes species, respectively (Fig 7 and S4C-S4E Figure in S1 File). Two less abundant species *m/z* 698 (PE a-34:3) and *m/z* 701 (PE a-34:1) also showed a significant decrease in about 90.2% and 87.4% over the WT and an average about 68% (for both Pes) over MFR strain (Fig 7 and S4A, S4B Figure in S1 File). There was one particular alteration in *m/z* 714 (PE 34:2) with a significant decrease (by 55%) only for the treated WT line (Fig 7 and S4F Figure in S1 File).

The lipid remodeling induced by the oxidative stress became even more evident when multiple PE species unusually produced, or commonly detected in small amounts, consistently increased at relatively high levels after DMMB-PDT. In both strains, 9 PE species were considerably increased compared to each corresponding untreated control. All of these species showed higher levels of relative abundance for the MFR strain than the WT strain. From these, *m/z* 760 (PE 38:7) and *m/z* 762 (PE 38:6) presented the greatest increase upon MFR DMMB-PDT group (by 53.9-fold and 55.9-fold, respectively), while WT DMMB-PDT enhanced in 14.3-fold and 8.3-fold regarding the same PE species (Fig 7 and S4K and S4L Figure in S1 File).

For WT DMMB-PDT, we found that *m/z* 775 (PE 38:0) and *m/z* 854 (PE 44:3) PE contents were 5.5 times higher than WT untreated control. However, MFR DMMB-PDT comprised

even greater amounts of such species (8.7-fold and 19.1-fold) (Fig 7 and S4I and S4Q Figure in S1 File). The same trend was also observed for the other species: The *m/z* 792 (PE 40:6), *m/z* 794 (PE 40:5) were increased by 10.1-fold (WT DMMB-PDT), 13.9-fold (MFR DMMB-PDT), 13.7-fold (WT DMMB-PDT) and 17.9-fold (MFR DMMB-PDT), respectively (Fig 7 and S4M and S4N Figure in S1 File). Whilst there has been detected nearly 9-fold increase for WT DMMB-PDT over 823 (PE 42:4), in MFR DMMB-PDT values shown were about 11 times higher after treatment (Fig 6 and S4O Figure in S1 File). However, *m/z* 822 (PE 42:5) was the only case in which the relative abundance of this molecular species was higher for the WT (by 6.9-fold) than for MFR (by 5.2-fold) after treatment compared to their own control. However, no significant differences were found between both (Fig 7 and S4P Figure in S1 File).

It is important to mention that 3 particular species significantly increased post-treatment only in the MFR line–*m/z* 742 (PE 36:2) (by 3.3-fold), *m/z* 744 (PE 36:1) (by 4.2-fold), *m/z* 764 (PE 38:5) (by 14.8-fold), even though an upward trend was also detected in the WT (Fig 7 and S4G, S4H and S4J Figure in S1 File).

Several alterations were detected over the pool of PC species by assessing precursor ion scanning *m/z* 184 in positive mode. Although various PC species demonstrated a tendency to increase in the WT DMMB-PDT group, only 2 of them were statistically significantly different from the WT untreated control–*m/z* 820 (PC a-40:6) and *m/z* 818 (PC 38:0), revealing an increase by 2.6-fold and 3.0-fold (Fig 7 and S5J and S5T Figure in S1 File). We also noted a few decreases in other different PC species with 2 alterations statistically significant–*m/z* 834 (PC 40:6) and *m/z* 832 (PC 40:7), with a reduction of 74% in both species. Interestingly, these are two of the most abundant PC in WT cell line (Fig 7 and S5B and S5C Figure in S1 File).

Remarkably, a different pattern between WT and MFR parasites was observed in response to DMMB-PDT over choline-containing PLs. We found that 9 PC species were significantly increased in MFR parasites, while 13 species showed a substantial decrease after DMMB-PDT. Interestingly, the most abundant PC species were substantially reduced after therapy, while the lower ones increased.

A 1.6-fold increase for *m/z* 803 (PC 38:7) and *m/z* 823 (PC a-40:4) was observed followed by 2.0-fold increase of *m/z* 800 (PC a-38:2), *m/z* 866 (PC 42:4) and *m/z* 867 (PC 42:3) species (Fig 7 and S5F, S5I, S5Y and S5E Figure in S1 File). The biggest changes were identified in 4 species–*m/z* 802 (PC a-38:1), *m/z* 820 (PC a-40:6), *m/z* 848 (PC a-42:7), *m/z* 818 (PC 38:0), showing a raise by 2.5-fold in these molecular species (Fig 7 and S5J, S5K and S5T Figure in S1 File).

Regarding the reduced PC species, *m/z* 760 (PC 34:1) presented the lowest decrease in abundance (28%), followed by *m/z* 734 (PC 32:0) (34%), *m/z* 772 (PC a-36:2) and *m/z* 790 (PC 36:0), both reduced in 38% (Fig 6 and S5D, S5L, S5M, S5P Figure in S1 File). PC species with *m/z* peaks of 788 (PC 36:1) and *m/z* 808 (PC 38:5) resulted in further decrease, on average by 50%, while *m/z* 811 (PC 38:4), *m/z* 839 (PC 40:3) and *m/z* 785 (PC 36:2) reduced in 55% (Fig 7 and S5Q, S5R, S5W, S5X and S5Z Figure in S1 File). The *m/z* 812 (PC 38:3) showed a 60% reduction, whereas *m/z* 837 (PC 40:4), *m/z* 834 (PC 40:6) and *m/z* 832 (PC 40:7) revealed the most significant decrease, 70% lower in abundance compared to untreated MFR parasites (Fig 7 and S5A-S5C, and S5V Figure in S1 File).

The PS contents showed a great decrease in all species in the WT strain, with significant alterations in 6 species–*m/z* 853 (PS 42:11) and *m/z* 903 (PS 44:1) were reduced by 98%, while *m/z* 855 (PS 42:10), *m/z* 880 (PS 44:12) were found in 95% lesser amounts than WT control (Fig 7 and S6D-S6F, S6H Figure in S1 File). In addition, we found that *m/z* 829 (PS 40:9) and *m/z* 801 (PS 38:9) PS species reduced in 82% and 89% (Fig 7 and S6B and S6C Figure in S1 File). In contrast, PS content in MFR parasites increased after DMMB-PDT, with 2 significant alterations–*m/z* 829 (PS 40:9) and *m/z* 892 (PS 44:6), resulting in abundance about 4 and 40 times higher than untreated parasites (Fig 7 and S6B and S6G Figure in S1 File).

## Discussion

The rise of antileishmanial drug resistance over the past decades has become one of the primary causes of leishmaniasis treatment failure [12]. In recent years, increasing attention has been drawn to the use of PDT as a potential alternative to treat CL to prevent the widespread number of recurrence and unresponsive cases [19].

In our study, we have demonstrated that DMMB-PDT was effective in the inactivation of both *L. amazonensis* promastigotes strains (Fig 2). The oxidative stress produced by DMMB-PDT played a significant role in parasites death by disturbing cellular homeostasis despite *Leishmania* antioxidant defenses.

Moreover, it has been previously reported that DMMB-PDT achieved similar results against the intracellular forms of the WT and MFR *L. amazonensis*. It has been shown that intracellular amastigotes were successfully inactivated on infected macrophages. Both strains were equally susceptible to DMMB-PDT regardless of the parasites (extracellular or intracellular) stages. Importantly, this was accomplished without producing significant toxicity to mammalian cells (fibroblasts and macrophages) [22].

One of the major advantages of PDT is the potential to generate different types of ROS either by type I or type II reactions. Type I process is given by the transfer of charge (from the excited photosensitizer) to a substrate to form a radical (anion or cation), which may further react with oxygen and produce ROS, including $H_2O_2$, hydroxyl radical, and superoxide anions [33]. Alternatively, the triplet state of the photosensitizer may undergo a type II reaction, which involves energy transfer to molecular oxygen, producing singlet oxygen, a highly reactive oxidizing agent [33, 34].

Both reactions are driven by the intrinsic characteristics of the photosensitizer, which in combination with a proper light source will determine the efficiency of the therapy [14]. In this regard, our scope was to verify if resistant and wild-type *Leishmania* cells could generate different levels of ROS. Curiously, the MFR-resistant strain presented higher ROS amounts than wild-type, although DMMB-PDT promoted pronounced killing for both strains at 750 nM and 50 J/cm$^2$. This finding indicates that resistant strains were not able to reduce oxidative stress induced by DMMB-PDT (see EC$_{50}$, Fig 2).

DMMB is advantageous for exhibiting a long visible wavelength light absorption in the red region of the spectrum, producing a quantum yield of singlet oxygen around 70% [34]. Moreover, it is unlikely to be reduced. Yet, the two additional methylene groups in DMMB make it highly lipophilic with a positive log P$_{o/w}$ (+ 1.01), thereby it easily diffuses across the lipid bilayer of the plasma membrane [35]. In addition, it has been shown that having a positive charge enables it to accumulate in the mitochondria (negatively charged inside) [34, 35]. The resulting irreversible mitochondrial depolarization jeopardizes oxidative phosphorylation and compromises adenosine triphosphate (ATP) synthesis. This impairment of mitochondrial respiration is a key factor involved in cell death [36].

Indeed, this is in agreement with our microscopic observations of mitochondrial labeling (Fig 4). The diffusion of MitoTracker staining throughout parasite cytoplasm suggests that a disruption in mitochondria structural integrity and a loss of its membrane potential occurred directly after DMMB-PDT. Because of the extensive photodamage, parasites failed to restore their mitochondrial function, thereby 1 h after irradiation no signs of the organelle were observed. Therefore, our results suggest that DMMB-PDT promoted a complete mitochondrial depolarization in both parasite strains as a consequence of low membrane potential.

Another key point addressed was the increase of LDs in both treated strains (Fig 5). Lipid droplets are storage organelles usually present in small numbers under physiological conditions [37]. They have long been known as energy reservoirs, mostly for neutral lipids such as

triacylglycerol and sterol esters. In recent years, they have been recognized as independent dynamic organelles also involved in cellular homeostasis [37]. Lipid droplets can act beyond lipid metabolism and interact with other organelles such as mitochondria, endoplasmic reticulum, peroxisomes, and therefore play a pivotal role in the management of redox imbalance [38].

LDs tend to accumulate under cellular stress conditions, protecting membranes from the peroxidation process. In the case of oxidative stress, the overproduction of ROS promotes rapid lipid peroxidation, which is quite toxic to the cells due to the excess of fatty acids (FA) and their derivatives released [38, 39]. In this way, lipid droplets can act as buffers and sequester and store the pool of free FA to prevent lipotoxicity, maintaining the redox state, and ensuring cellular survival [38]. The lipids stored can further be used as an energy source and supply gradually the FA necessary for posterior phospholipids synthesis [37].

Therefore, our results suggest that the high yield of ROS generated by DMMB-PDT promoted a significant lipid peroxidation in cellular membranes, especially in the mitochondria, which led to a loss of membrane potential and releasing of lipids to the cytosol. As a response to metabolic stress, LDs substantially increased, accumulating large amounts of free lipids as an attempt to maintain homeostasis in the parasites. Indeed, the association of LDs and mitochondrial metabolism are essential to prevent lipotoxicity and maintain parasite viability [38]. However, due to the lipid overload there was a dysfunction in LDs and they probably failed to regulate lipotoxicity and redox imbalance, which led to constant cellular stress resulting in parasite death.

To understand in more detail the role of lipid metabolism in *L. amazonensis*, we performed a comprehensive quantitative lipidomics analysis of both WT and MFR strains. Firstly, we compared the differences in lipid profile between both lines, and then we further evaluated their phospholipid alterations under oxidative stress conditions.

ES-MS-MS spectra demonstrated several differences over the lipid metabolism between both strains, which could primarily explain the mechanism of resistance to miltefosine in MFR *L. amazonensis*. The miltefosine resistance process has been related to a large reduction of drug internalization resulting from a defect in miltefosine transporter (MT) and its regulatory non-catalytic subunit Ros3 [24]. Because of an impaired transport function, miltefosine can be expelled by cells even in high drug concentrations [23]. In fact, these genes are responsible for an inward translocation of miltefosine and PLs throughout the plasma membrane [23, 24]. However, it has been previously reported that the MFR phenotype used in our study had a single point of mutation only in the MT gene, rather than both MT and Ros3 [24]. In addition, there is growing evidence that MF resistance could also be correlated with other factors such as PL composition, FA and sterols metabolism [40–42].

It has been reported that miltefosine-resistant *Leishmania donovani* species presented increased levels of PC content [43]. Likewise, our results demonstrate that the overall abundance of this PL class was substantially higher in the MFR strain compared to the WT counterpart. PC is the most abundant phospholipid class in eukaryotic membranes and its biosynthesis occurs via the CDP-choline pathway, one of the two branches of the Kennedy pathway [44]. Briefly, choline is imported by cells and phosphorylated to phosphocholine via choline kinase. A second reaction involves the production of CDP-choline (via CTP:phosphocholine-cytidylyltransferase) followed by a final step, which is the formation of PC [45].

Miltefosine belongs to the class of alkylphosphocholine drugs; therefore it acts as a phosphocholine analog, inhibiting the translocation of CTP:phosphocholine-cytidylyltransferase, and consequently PC synthesis [46]. Thus, the upregulation of PC observed in the MFR phenotype suggests an adaptive mechanism of parasites to overcome the inhibition of the CDP-choline pathway and promote their survival.

The second branch of the Kennedy pathway relies on the synthesis of PE, via the CDP-etha-nolamine pathway [44, 45]. PE is the second most abundant PL class in *Leishmania* parasites and it is closely related to PC synthesis [30]. In fact, PC can be synthesized either by CDP-cho-line or by the conversion of PE to PC [44]. The alternative route for the synthesis of PC involves threefold methylation of PE, which is catalyzed by phosphatidylethanolamine N-methyltransferase [47]. Thus, we believe that the PE N-methylation pathway could be a possi-ble reason for the significant decrease found in one of the most abundant PE species (PE a-36:2) upon MFR parasites.

Apart from this PE species, no further alterations were observed between MFR and WT phenotypes upon the overall abundance of PE. However, we assume that PE levels were main-tained, because of PS decarboxylation, an alternative route used for PE biosynthesis. As *Leish-mania* possesses orthologs of phosphatidylserine synthase and phosphatidylserine decarboxylase, PE could have been also generated from PS [47]. Since PS was found in lower abundance on MFR parasites compared to the WT, we hypothesize this pathway could be involved in such phospholipids alterations.

The analysis of inositol-containing phospholipids also revealed significant changes over some PI species in the MFR strain. One possible reason could be related to the miltefosine mode of action, which involves the inhibition of phosphatidylinositol 3-kinase (PI3K)/protein kinase B (akt or PKB) pathway, which is essential for signal transduction and cell survival [23, 48]. PI3K is a key protein responsible for phosphoinositides formation [48], thus the overpro-duction of PI species may have a role in the molecular mechanism underlying miltefosine resistance developed by MFR parasites.

Alterations in phospholipids due to oxidative stress were also observed in both pheno-types. The extensive lipid peroxidation mediated by DMMB-PDT showed a rapid lipid remodeling as a consequence of significant damage in the parasite's membrane. It is impor-tant to note that lipid peroxidation may cause severe changes in the lipid bilayer altering membrane permeability, fluidity, and structural integrity, thereby resulting in a subsequent cellular dysfunction [49].

As previously mentioned, our findings suggested an important mitochondrial oxidative damage, which is consistent with lipid alterations detected by ES-MS-MS. Cardiolipin is a mitochondria-exclusive phospholipid constituent, and its biosynthesis is related to PA and PG metabolism [50]. Initially, PA is converted to cytidinediphosphate-diacylglycerol (CDP-DAG), which is the precursor of PG. PG will then generate cardiolipin by the addition of another CDP-DAG molecule to PG [51]. PA and PG are minor PL classes present in *Leishmania* para-sites membrane, however, tandem mass spectra revealed relevant changes over some species. Precursor ions scanning for m/z 153 in the negative mode pointed to an expressive reduction in PA species while PG species were greatly upregulated. Collectively, these data suggest that changes in the metabolism of these lipids may have played a role in cardiolipin synthesis to overcome mitochondrial membrane damage.

Nevertheless, the overall PG synthesis was significantly greater in the WT strain than in the MFR phenotype. As a key molecule, PA can be either converted into CDP-DAG to form PI, PG, and cardiolipin, or it can be used as a precursor for DAG to form PC, PE, and PS [30]. Since MFR parasites are dependent upon high levels of PC, it is possible that PA was more involved in DAG pathway to form PC, rather than PG.

Drug-induced lipid alterations have been reported in several *Leishmania* species, mostly over PC and PE metabolism [31, 42]. Indeed, the biggest lipid changes observed in our study were related to PC/PE classes, in which the most abundant species were significantly downre-gulated in both phenotypes. As these are the main PL classes containing polyunsaturated fatty acids, they are more susceptible to ROS, therefore prone to oxidation [49]. Alternatively,

parasites underwent a rapid lipid remodeling in response to oxidative stress, producing other molecular species not usually detected under physiological conditions. Of note, the overproduction of PC was more pronounced in MFR phenotype compared to the WT, probably due to their higher demand for this lipid class.

Interestingly, unlike in mammalian cells, most PE species in *Leishmania* are displayed in the form of plasmalogens, which in turn, should act as membrane antioxidants scavenging radical species [30, 44]. However, those species were also significantly decreased, whereas the minor ones were greatly upregulated, suggesting that the amount of ROS produced was higher than the ability of the parasites to reduce oxidative stress. Differences between PE for MFR and WT strains could explain the higher amount of ROS observed for MFR strain following DMMB-PDT (see Figs 3 and 6).

Surprisingly, those upregulated PE species comprised a long chain of fatty acids, which was also observed over the others PL classes. Indeed, it has been shown that the increase in the chain of fatty acids in *Leishmania* may have an important role underlying the cellular mechanism of protection, which has been previously reported for *Leishmania donovani* [52]. Long carbon chains enhance membrane rigidity and reduce its fluid properties, thereby preventing drug permeability into the plasma membrane [31]. In addition, it is believed that longer fatty acids can be used as an energy source in parasites [31]. Therefore, our results suggest that the synthesis of phospholipids containing long FA chains could have been a strategy developed by both *L. amazonensis* strains to help promote their survival.

The role of PE over *Trypanosomatidae* is considerably critical, because some parasite species are significantly sensitive to the decrease of PE levels, particularly upon the mitochondrial inner membrane [44, 53]. Thus, we suggest that to compensate for the loss of PE content, levels of PS were also modulated after DMMB-PDT via PS synthase and/or decarboxylase. However, both lines showed a different response under stress conditions. While PS amounts were substantially reduced in the WT parasites, MFR had some species increased.

PS is synthesized in the endoplasmic reticulum and is transported to mitochondria to produce PE [54]. Since levels of PE were decreased in the WT cells after DMMB-PDT, we assume PS species were rapidly decarboxylated in mitochondria to form PE (via PS decarboxylase). On the other hand, as the relative abundance of PS was particularly low under normal conditions in the MFR phenotype, we hypothesize that PS synthesis (via PS synthase) was required to improve PE levels in mitochondria membrane upon oxidative stress.

Perturbations in PE content affect not only phospholipids in the membrane, but also result in protein dysfunction, especially in those lipid-interacting proteins [53]. For example, glycosylphosphatidylinositol (GPI)-anchored proteins are inserted in the outer layer of the plasma membrane and are essential for parasite virulence and survival [55]. They are involved in membrane surface protection and cellular nutrition, and their biosynthesis relies on the addition of ethanolamine phosphate groups to PI (together with monosaccharides). It has been shown that levels of GPI are dependent on the synthesis of inositol in *T. brucei* bloodstream form [56]. Thus, changes in ethanolamine levels might also affect the modulation of PI as a consequence of damage in GPI anchors [57]. Indeed, our results suggest that DMMB-PDT leads to increased levels of some PI species as a result of changes in PE metabolism caused by the perturbations produced in the plasma membrane.

In conclusion, our results demonstrate that DMMB-PDT killed both wild-type and miltefosine resistant strains effectively, promoting mitochondrial dysfunction via loss of membrane potential and lipid droplet accumulation with significant alterations of lipid metabolism. Moreover, the MFR line was found to be more susceptible to oxidative stress than the WT strain, resulting in increased levels of ROS at low nanomolar DMMB concentrations. We hope these data will encourage other studies as this is the first report on lipidomics of miltefosine-

resistant and susceptible *L. amazonensis* parasites following oxidative stress. For the time being, DMMB-PDT appears to be a promising strategy for overcoming the challenges of antil-eishmanial drug resistance in CL infections.

## Supporting information

**S1 File.**
(DOCX)

## Author Contributions

**Conceptualization:** Martha S. Ribeiro, Terry K. Smith.

**Formal analysis:** Michela Cerone, Saydulla Persheyev, Cheng Lian.

**Funding acquisition:** Ifor D. W. Samuel.

**Investigation:** Fernanda V. Cabral, Michela Cerone, Terry K. Smith.

**Supervision:** Ifor D. W. Samuel, Martha S. Ribeiro, Terry K. Smith.

**Writing – original draft:** Fernanda V. Cabral, Ifor D. W. Samuel, Martha S. Ribeiro, Terry K. Smith.

**Writing – review & editing:** Saydulla Persheyev, Cheng Lian, Ifor D. W. Samuel, Martha S. Ribeiro, Terry K. Smith.

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
