## [Decision Letter · Decision Letter 0]

23 Mar 2023

PONE-D-23-03795Photodynamic therapy as a promising strategy to overcome drug resistance in cutaneous leishmaniasisPLOS ONE

Dear Dr. Smith,

Thank you for submitting your manuscript to PLOS ONE. After careful consideration, we feel that it has merit but does not fully meet PLOS ONE’s publication criteria as it currently stands. Therefore, we invite you to submit a revised version of the manuscript that addresses the points raised during the review process.

We look forward to receiving your revised manuscript.

Kind regards,

Mehdi Bamorovat

Academic Editor

PLOS ONE

Journal Requirements:

"We thank Prof. Silvia Reni Bortolin Uliana from University of São Paulo 

for providing the WT and MFR L. amazonensis strains. We also thank 

Photonics Institute (INFO) from Conselho Nacional de Desenvolvimento 

Científico e Tecnologico (CNPq, grant #465763/2014-6), and Comissão 

Nacional de Energia Nuclear (CNEN) for financial support. F.V. Cabral thanks 

CNPq and Coordenação de Aperfeiçoamento de Pessoal de Nível Superior

(CAPES, #88887.364974/2019-00) for her scholarships. We are also grateful 

to the Royal Society (grant CH160144) and the Scottish Funding Council (grant 

xxx) for financial support."

"No 

Note that wild type rabies may also be classed at Cat 2 rather than Cat 3 under the HSE if appropriate safety requirements are met (based on https://www.hse.gov.uk/pubns/misc208.pdf )"

Reviewers' comments:

Reviewer's Responses to Questions

**Comments to the Author**

1. Is the manuscript technically sound, and do the data support the conclusions?

Reviewer #1: Yes

Reviewer #2: Partly

2. Has the statistical analysis been performed appropriately and rigorously? 

Reviewer #1: Yes

Reviewer #2: Yes

3. Have the authors made all data underlying the findings in their manuscript fully available?

Reviewer #1: Yes

Reviewer #2: Yes

4. Is the manuscript presented in an intelligible fashion and written in standard English?

Reviewer #1: Yes

Reviewer #2: Yes

5. Review Comments to the Author

Reviewer #1: General comments

The authors have performed an in vitro study to evaluate the activity of photodynamic therapy (PDT) using the wild-type and a lab-induced miltefosine-resistant strain. The work is interesting and extensive; however, this in vitro experimental study has been done on the non-clinical stage (promastigotes) which is in the gut of the female sandflies and also the culture media. The story can be different when dealing with the clinical stage (amastigotes or Leishman bodies) which is resided inside the macrophage of humans and animal hosts. These two stages are significantly different in terms of physiology and also molecular makeup.

The study needs clarification and modifications:

- The light source should be specified.

- The distance between the light source and the target cells in PDT should be clarified

- The time length used in PDT should be mentioned.

- Specify the exact amount of ROS production in PDT.

- Present fluorescent peaks and their intensity using BMMB.

-

Reviewer #2: The manuscript “Photodynamic therapy as a promising strategy to overcome drug resistance in cutaneous leishmaniasis” evaluated the activity of photodynamic therapy (PDT) using the photosensitizer 1,9-dimethyl methylene blue (DMMB), against a wild-type (WT) and a lab induced miltefosine-resistant (MFR) Leishmania amazonensis strains in vitro.

1. The manuscript is well presented but there are major concerns that should be addressed by the authors. The title should be clear about the in vitro approach of the study, and there are many questions to be answered before to define the PDT-DMMB as a promising strategy of cutaneous leishmaniasis treatment. I highly suggest the authors to modify the title to make it in line with the results and conclusions presented.

2. There was no study of cytotoxicity in mammal cells. As the PDT-DMMB induces ROS production in a nonspecific way, the evaluation of the treatment in host cells is crucial to ensure the safety of the treatment.

3. The study was carry out in promastigote forms only. An efficacy study should also be performed against intracellular amastigote forms, since this is the form present in the cells of the hosts that will be submitted to the treatment.

Find below other points that should be addressed by the authors.

• Please keep the results description objective, and avoid the use of subjective sentence as “very”.

• In the last sentence of abstract, please revise the use of the word “treat”, since the goal is not to treat but to kill Leishmania parasite.

• Please, give details about the light source, distance from the light source to the bottom plate. Was the irradiation direct or on the plate cover?

• Was the dilution factor of miltefosine and DMMB 1:2?

• Why did immunofluorescence microscopy use 8 J/cm2 while the other assays used 50 J/cm2?

• In lipid analysis, please detail DMMB-PDT treatment.

• In Figure 1B, was the EC50 value lower than the low DMMB concentration evaluated? If yes, what was the R2 value for this? Would it appropriate to adequate evaluate lower concentrations?

• The figure 1 legend missed information, such as miltefosine treatment concentration in fig 1B, DMMB in fig 1A. please revise the legends and add these and other important information.

• In figure 2, the x-axis title for the different treatment is confuse, please revise. Revise the legend as well.

• It is confuse to follow the different groups of treatment since there is no consistent in the group’s name. Figure 3 caption and legends in the image is very confuse, per example. The authors should be consistent in the denomination of the groups and in the use of terms such as DMMB-PDT throughout the manuscript.

• Please review and alter legend and caption of figure 3.

• The results of mitochondria membrane potential and lipid droplets are presented in the Figures 3 and 4. Can the authors to quantify the fluorescence area in these and other images and present qualitative data that should be statistical evaluated?

• The least sentence of the second paragraph of discussion it is confuse and should be reviewed.

• The lack of number line make it difficult to reference text during revision.

6. PLOS authors have the option to publish the peer review history of their article (what does this mean?). If published, this will include your full peer review and any attached files.

Reviewer #1: **Yes: **Iraj Sharifi

Reviewer #2: **Yes: **Fernando Almeida-Souza

---

## [Author Response · Author response to Decision Letter 0]

16 May 2023

Response to Reviewers

Dear reviewer #1,

Thank you very much for your time and consideration of our manuscript. Below you will find our responses to your comments. All changes are highlighted in yellow in the manuscript.

The authors have performed an in vitro study to evaluate the activity of photodynamic therapy (PDT) using the wild-type and a lab-induced miltefosine-resistant strain. The work is interesting and extensive; however, this in vitro experimental study has been done on the non-clinical stage (promastigotes) which is in the gut of the female sandflies and also the culture media. The story can be different when dealing with the clinical stage (amastigotes or Leishman bodies) which is resided inside the macrophage of humans and animal hosts. These two stages are significantly different in terms of physiology and also molecular makeup. The study needs clarification and modifications:

R. The reviewer is right and the two forms differ physiologically and molecularly. However, the main purpose of this work was to demonstrate the possible mechanisms of action of PDT using DMMB as a photosensitizer focusing on the changes in the lipid metabolism of Leishmania amazonensis, including a miltefosine-resistant strain. As far as we know, there are no reports about the lipidomic analysis of L. amazonensis miltefosine-resistant strains. Besides, this is the first attempt to understand the lipidomics of Leishmania parasites after PDT. We believe this advance will be of interest to a wide range of researchers working on parasitic diseases, including those interested in PDT. However, we understand that further studies are necessary and addressed this issue in the manuscript.

- The light source should be specified.

R. We apologize for that. We used a red LED (660 nm peak wavelength, 25 nm full width at half maximum). We included the information in the text. 

- The distance between the light source and the target cells in PDT should be clarified.

R. The light source illuminates the wells from underneath, with a distance of 4 mm from the bottom of the well. We included this information in the text.

- The time length used in PDT should be mentioned.

R. Parasites were irradiated using a red LED at an irradiance of 20 mW/cm2 for 2500 s to deliver a dose of 50 J/cm2. We included the information in the text. 

- Specify the exact amount of ROS production in PDT.

R. ROS production was measured by fluorescence using the indicator 2'-7'-dichlorodihydrofluorescein diacetate (DCFH-DA), a dye whose fluorescence is proportionally related to ROS production in live cells. We found that after DMMB-PDT the WT strain produced an average fluorescence intensity of 38410±4200 a.u., while the resistant strain (MFR) produced an average of 50489±10856 a.u., i.e., the MFR strain generates about 24% more ROS than WT.

- Present fluorescent peaks and their intensity using BMMB.

R. We added a figure to show the spectra of DMMB. 

Dear reviewer #2,

Thank you very much for your time and consideration of our manuscript. Below you will find our responses to your comments. All changes are highlighted in yellow in the manuscript.

1. The manuscript is well presented but there are major concerns that should be addressed by the authors. The title should be clear about the in vitro approach of the study, and there are many questions to be answered before to define the PDT-DMMB as a promising strategy of cutaneous leishmaniasis treatment. I highly suggest the authors to modify the title to make it in line with the results and conclusions presented.

R. The reviewer is right and we thank your suggestions. We changed the title of the study to “New insights in photodynamic inactivation of Leishmania amazonensis: a focus on lipidomics and resistance" and addressed this point in the text to make clear the aim of the paper.

2. There was no study of cytotoxicity in mammal cells. As the PDT-DMMB induces ROS production in a nonspecific way, the evaluation of the treatment in host cells is crucial to ensure the safety of the treatment.

R. Indeed, the cytotoxic effects of PDT mediated by DMMB have been addressed in our previous studies (Cabral, F. V. Towards new therapeutic strategies for cutaneous leishmaniasis. text, University of Sao Paulo, 2021. https://www.teses.usp.br/teses/disponiveis/85/85134/tde-18112021-095721/pt-br.php). We found that PDT was not toxic to fibroblasts or macrophages even at higher light doses or DMMB concentrations. To clarify that, we included this reference in the text.

3. The study was carry out in promastigote forms only. An efficacy study should also be performed against intracellular amastigote forms, since this is the form present in the cells of the hosts that will be submitted to the treatment.

R. Indeed, the potential of DMMB-PDT on intracellular amastigotes has also been demonstrated in our previous studies (Cabral, F. V. Towards new therapeutic strategies for cutaneous leishmaniasis. text, University of Sao Paulo, 2021. https://www.teses.usp.br/teses/disponiveis/85/85134/tde-18112021-095721/pt-br.php). However, the present work aimed to demonstrate new insights into the possible PDT mechanisms and a comprehensive study on lipidomics of a WT and a miltefosine-resistant strain of Leishmania parasites before and after PDT. We understand that further studies are necessary, however, we believe our data open an avenue to help combat drug resistance in cutaneous leishmaniasis. To clarify that, we addressed this issue in the text.

Find below other points that should be addressed by the authors.

• Please keep the results description objective, and avoid the use of subjective sentence as “very”.

R. Thank you for your suggestion. We carefully revised the description of the results.

• In the last sentence of abstract, please revise the use of the word “treat”, since the goal is not to treat but to kill Leishmania parasite.

R. We have changed the word “treat” to “kill” as suggested.

• Please, give details about the light source, distance from the light source to the bottom plate. Was the irradiation direct or on the plate cover?

R. We apologize for that. We used a red LED (660 nm peak wavelength, 25 nm full width at half maximum). The experimental setup illuminates the wells from underneath, with a distance of 4 mm from the bottom of the well. We included the information in the text. 

• Was the dilution factor of miltefosine and DMMB 1:2?

R. Yes, the dilution factor for both compounds was 1:2. We added this information to the manuscript.

• Why did immunofluorescence microscopy use 8 J/cm2 while the other assays used 50 J/cm2?

R. Indeed, we used immunofluorescence to understand the role of PDT on parasites. In this case, reducing the light dose to a sublethal dose is more appropriate. We clarified this issue in the manuscript. 

• In lipid analysis, please detail DMMB-PDT treatment.

R. We added information about DMMB-PDT in the text as per your request. 

• In Figure 1B, was the EC50 value lower than the low DMMB concentration evaluated? If yes, what was the R2 value for this? Would it appropriate to adequate evaluate lower concentrations?

R. The R2 found for DMMB-PDT dose-response curve was 0.98 for both WT and MFR. We added this information to the manuscript. Indeed, we obtained the EC50 values to demonstrate the resistance to miltefosine and the susceptibility of MFR strains to PDT. To conduct the experiments, we used the conditions established in our previous manuscript (Cabral, F. V. et al. Organic Light‐Emitting Diodes as an Innovative Approach for Treating Cutaneous Leishmaniasis. Adv Mat Technol 2021, 2100395). 

• The figure 1 legend missed information, such as miltefosine treatment concentration in fig 1B, DMMB in fig 1A. please revise the legends and add these and other important information.

R. We added the information about miltefosine treatment in the figure 1. 

• In figure 2, the x-axis title for the different treatment is confuse, please revise. Revise the legend as well.

R. We revised the x-axis title and legend.

• It is confuse to follow the different groups of treatment since there is no consistent in the group’s name. Figure 3 caption and legends in the image is very confuse, per example. The authors should be consistent in the denomination of the groups and in the use of terms such as DMMB-PDT throughout the manuscript.

R. We revised carefully the text to maintain coherence.

• Please review and alter legend and caption of figure 3.

R. We revised and changed the legend.

• The results of mitochondria membrane potential and lipid droplets are presented in the Figures 3 and 4. Can the authors to quantify the fluorescence area in these and other images and present qualitative data that should be statistical evaluated?

R. Indeed, statistical analysis for these assays would not add new information. Our purpose was to demonstrate the impact of PDT on both Leishmania strains compared to untreated controls. The images clearly show that no differences are noticed comparing both strains. 

• The least sentence of the second paragraph of discussion it is confuse and should be reviewed.

R. We revised the sentence as per your request.

• The lack of number line make it difficult to reference text during revision.

R. We added line numbers in the text.

---

## [Decision Letter · Decision Letter 1]

8 Jun 2023

PONE-D-23-03795R1New insights in photodynamic inactivation of Leishmania amazonensis: a focus on lipidomics and resistancePLOS ONE

Dear Dr. Smith,

Thank you for submitting your manuscript to PLOS ONE. After careful consideration, we feel that it has merit but does not fully meet PLOS ONE’s publication criteria as it currently stands. Therefore, we invite you to submit a revised version of the manuscript that addresses the points raised during the review process.

We look forward to receiving your revised manuscript.

Kind regards,

Mehdi Bamorovat

Academic Editor

PLOS ONE

Journal Requirements:

Reviewers' comments:

Reviewer's Responses to Questions

**Comments to the Author**

1. If the authors have adequately addressed your comments raised in a previous round of review and you feel that this manuscript is now acceptable for publication, you may indicate that here to bypass the “Comments to the Author” section, enter your conflict of interest statement in the “Confidential to Editor” section, and submit your "Accept" recommendation.

Reviewer #1: All comments have been addressed

Reviewer #2: All comments have been addressed

2. Is the manuscript technically sound, and do the data support the conclusions?

Reviewer #1: Yes

Reviewer #2: Yes

3. Has the statistical analysis been performed appropriately and rigorously? 

Reviewer #1: Yes

Reviewer #2: Yes

4. Have the authors made all data underlying the findings in their manuscript fully available?

Reviewer #1: Yes

Reviewer #2: Yes

5. Is the manuscript presented in an intelligible fashion and written in standard English?

Reviewer #1: Yes

Reviewer #2: Yes

6. Review Comments to the Author

Reviewer #1: Comments

The authors evaluated the activity of photodynamic therapy (PDT) in vitro employing DMMB against two strains of Leishmania amazomensis; the wild-type (WT) and a lab-induced miltefosine-resistant (MFR) phenotype. Although the study is interesting and extensive some questions remain to be answered and cleared before this paper is considered for publication.

1-Do the author mean by "overcome drug resistance in cutaneous leishmaniasis" the use of photodynamic therapy as an adjunct to drug therapy, or is it suggested as a separate solution without the presence of drugs?

2-What does it mean by "they absorb light in the red region of the spectrum, and red light penetrates deeper into tissue than shorter wavelengths"? Are the wounds caused by L. amazonensis not on the surface of the skin and is there a need for the presence of light in deeper tissues?

3- It is suggested to measure and investigate the amount of reactive singlet oxygen (1O2) created by the presence of light-sensitive material. In this case, you can refer to the article with doi: 10.1016/j.bpj.2011.12.043.

4- More details of photodynamic therapy conditions need to be explained.

5- In the discussion of the text of the article, the presence of ROS in light-sensitive species is described, but no information is given about the calculation of these species separately.

6- The author should write this work has been done using non-clinical promastigote (an extracellular stage in sandflies and culture media). The story could be different using the intramacrophage amastigote assay before conducting the study in advanced in vivo and clinical models.

Reviewer #2: In the second version of the manuscript, the authors addressed all my concerns. However, although the new title seems more appropriate now, the authors still need to clarify the in vitro approach of the study in the title of the manuscript.

7. PLOS authors have the option to publish the peer review history of their article (what does this mean?). If published, this will include your full peer review and any attached files.

Reviewer #1: No

Reviewer #2: **Yes: **Fernando Almeida-Souza

---

## [Author Response · Author response to Decision Letter 1]

19 Jul 2023

Response to Reviewers

Dear reviewer #1,

Thank you very much for your time and consideration of our manuscript. Below you will find our responses to your comments. All changes are highlighted in yellow in the manuscript.

1 - Do the author mean by "overcome drug resistance in cutaneous leishmaniasis" the use of photodynamic therapy as an adjunct to drug therapy, or is it suggested as a separate solution without the presence of drugs?

R. Photodynamic therapy (PDT) is an attractive light-based technology that involves the use of a photosensitizer and light to generate reactive oxygen species that can kill microorganisms. Because of its multi-target features, PDT can attack simultaneously lipids, proteins, and nucleic acids. Because of this characteristic PDT is unlikely to produce resistance, as opposed to the current drugs available to treat Leishmaniasis. In addition, PDT has been shown to promote antimicrobial activity against several pathogens, including multidrug-resistant bacteria and some yeast species. However, the use of PDT on drug-resistant Leishmania remains unexplored. We intended to show that PDT can effectively kill both the wild-type and a drug-resistant strain, i.e. miltefosine-resistant. Thus, we propose that PDT can overcome the growing drug resistance problem in cutaneous leishmaniasis by using topically the PS and delivering light directly to the target. We addressed this issue in the manuscript.

2 - What does it mean by "they absorb light in the red region of the spectrum, and red light penetrates deeper into tissue than shorter wavelengths"? Are the wounds caused by L. amazonensis not on the surface of the skin and is there a need for the presence of light in deeper tissues?

R. Indeed, light is absorbed by the photosensitizer. However, the red light that is not absorbed may be scattered forward and penetrate deeper into biological tissues. We rephrased the sentence for better/clearer understanding. 

3- It is suggested to measure and investigate the amount of reactive singlet oxygen (1O2) created by the presence of light-sensitive material. In this case, you can refer to the article with doi: 10.1016/j.bpj.2011.12.043.

R. Indeed, our purpose was not to correlate singlet oxygen generation with PDT dosimetry as Jarvi and collaborators. Here, we intended to verify if resistant Leishmania parasites were susceptible to high levels of ROS generated inside the cells, that they were unable to deal with, subsequently causing cell death

4- More details of photodynamic therapy conditions need to be explained.

R. We have added a lot more detail, including light doses and exposure times to the section “Light source and photosensitizer”.

5- In the discussion of the text of the article, the presence of ROS in light-sensitive species is described, but no information is given about the calculation of these species separately.

R. The calculation of ROS was given by the values obtained using the indicator DCFH, which is directly proportional to the total ROS production. However, the DCFH indicator can only measure the total amounts of ROS produced by PDT. Thus, we cannot address which ROS species are being generated at that time point evaluated. As above mentioned, our purpose was only to verify if resistant Leishmania parasites were equally susceptible to oxidative stress as the wild-type. Curiously, resistant cells were more susceptible to ROS than wild type. This result indicates that resistant strains were not able to reduce oxidative stress. Indeed, differences in phosphatidylethanolamine content between strains could explain these data. We addressed this issue in the manuscript.

6- The author should write this work has been done using non-clinical promastigote (an extracellular stage in sandflies and culture media). The story could be different using the intramacrophage amastigote assay before conducting the study in advanced in vivo and clinical models.

R. We added “Leishmania promastigotes” to the keywords to clarify this issue. However, the potential of DMMB-PDT on intracellular amastigotes has also been demonstrated in our previous studies (Cabral, F. V. Towards new therapeutic strategies for cutaneous leishmaniasis. text, University of Sao Paulo, 2021. https://www.teses.usp.br/teses/disponiveis/85/85134/tde-18112021-095721/pt-br.php). In addition, the mechanisms and a comprehensive study on lipidomics of a WT and a miltefosine-resistant strain of Leishmania parasites before and after PDT have still not been reported and this is one of the purposes of this manuscript. We understand that further studies are necessary, but we believe our data here opens novel avenues to help combat drug resistance in cutaneous leishmaniasis and goes someway to an initial understanding of how/why PDT is of therapeutic value against Leishmania.

Dear reviewer #2,

Thank you again for your time and consideration of our manuscript. Below you will find our responses to your comment. The change is highlighted in yellow in the manuscript.

In the second version of the manuscript, the authors addressed all my concerns. However, although the new title seems more appropriate now, the authors still need to clarify the in vitro approach of the study in the title of the manuscript.

R.. We understand that the title and abstract should highlight the scope of the paper and the abstract clearly mentions promastigotes and set as an in vitro study. Thus we have added Leishmania promastigotes to the keywords.

---

## [Editor Report · Decision Letter 2]

20 Jul 2023

New insights in photodynamic inactivation of Leishmania amazonensis: a focus on lipidomics and resistance

PONE-D-23-03795R2

Dear Dr. Smith

We’re pleased to inform you that your manuscript has been judged scientifically suitable for publication and will be formally accepted for publication once it meets all outstanding technical requirements.

Kind regards,

Mehdi Bamorovat

Academic Editor

PLOS ONE

---

## [Editor Report · Acceptance letter]

7 Sep 2023

PONE-D-23-03795R2 

New insights in photodynamic inactivation of *Leishmania amazonensis*: a focus on lipidomics and resistance 

Dear Dr. Smith:

I'm pleased to inform you that your manuscript has been deemed suitable for publication in PLOS ONE. Congratulations! Your manuscript is now with our production department. 

Kind regards, 

on behalf of

Dr. Mehdi Bamorovat 

Academic Editor

PLOS ONE